# Application of Fiber Biochar–MOF Matrix Composites in Electrochemical Energy Storage

**DOI:** 10.3390/polym14122419

**Published:** 2022-06-15

**Authors:** Meixiang Gao, Meng Lu, Xia Zhang, Zhenhui Luo, Jiaqi Xiao

**Affiliations:** 1State Key Laboratory of Biobased Material and Green Papermaking, Qilu University of Technology (Shandong Academy of Sciences), Jinan 250353, China; mxgao207@163.com (M.G.); L17864099213@163.com (M.L.); zx21619@163.com (X.Z.); 2Advanced Research Institute for Multidisciplinary Science, Qilu University of Technology (Shandong Academy of Sciences), Jinan 250353, China; 3School of Chemistry and Chemical Engineering, Qilu University of Technology (Shandong Academy of Sciences), Jinan 250353, China

**Keywords:** MOFs, biochar, ionic liquid method, supercapacitors, electrochemical properties

## Abstract

Fiber biochar–metal organic framework (MOF) composites were successfully prepared by three different biochar preparation methods, namely, the ionic liquid method, the pyrolysis method, and the direct composite method. The effects of the different preparation methods of fiber biochar on the physical and chemical properties of the biochar–MOF composites showed that the composite prepared by the ionic liquid method with the Zeolite-type imidazolate skeleton -67 (ZIF-67) composite after high temperature treatment exhibited a better microstructure. Electrochemical tests showed that it had good specific capacity, a fast charge diffusion rate, and a relatively good electrochemical performance. The maximum specific capacity of the composite was 63.54 F/g when the current density was 0.01 A/g in 1 mol/L KCl solution. This work explored the preparation methods of fiber biochar–MOF composites and their application in the electrochemical field and detailed the relationship between the preparation methods of the composites and the electrochemical properties of the electrode materials.

## 1. Introduction

The rapid development of the economy and technology has increased the demand for energy. The rapid consumption of traditional energy urgently requires us to explore sustainable and reliable energy storage in order to alleviate the problem of an energy shortage [1]. At present, there is a large demand gap for high-efficiency energy storage equipment (batteries, capacitors) in application, which has attracted the strong interest of researchers in the field of sustainable and reliable energy storage [2]. Supercapacitors are considered to be the most advantageous energy storage device. Compared with the traditional charging and discharging of batteries, supercapacitors have the advantage of faster charging and discharging rates [3], are environmentally friendly [4] have a higher rate density, and have a longer cycle life [5]. As a new type of energy storage container, supercapacitors can be widely used in portable electronic equipment [6], mobile equipment, and as a standby power supply.

MOFs, as a new type of porous material, have excellent characteristics of high porosity [7,8], large specific surfaces [9], and surface structure modifications [10]. In recent years, the synthesis and application of MOFs have attracted more and more attention [11,12,13]. ZIF-67 is one of the typical materials of cobalt-based MOFs, with an adjustable pore size structure and good thermal stability [14].

Most of the biomass waste (such as corn straw and sludge) produced each year is stacked or burned, resulting in ineffective utilization, which causes environmental pollution [15]. Fiber biochar is a product obtained by the chemical modification of biomass [16]. Biochar prepared from waste biomass can provide a feasible strategy for biomass waste management. It also helps to solve the problem of environmental pollution caused by burning biomass waste.

Biochar is widely used in adsorption, catalysis, energy storage, and other fields [17,18]. The precursors of biochar are varied with a relatively low price and rich source, and they are easy to obtain. It is environmentally friendly and easy to process and structurally design [19]. Biochar has gained significant attention not only to mitigate environmental concerns but also to promote the development of sustainable energy storage applications. However, its small specific surface area limits its application in electrochemical energy storage. Most researchers have used rice husk to prepare rice biochar. The modification methods have a direct influence on the electrochemical performance of the rice biochar as supercapacitor electrodes [20]. A single biochar is rarely applied in a supercapacitor; so, a relatively simple process was adopted in this work where biochar was compounded with other materials to improve its electrochemical performance. Other materials include organic semiconductors, inorganic metal complexes, organic metal complexes, etc. MOFs display excellent properties such as high porosity, an adjustable pore structure, and a large specific surface area; MOFs can be applied to the fields of gas separation and storage [21,22], catalysis [23], and electrochemical energy storage [24]. However, MOFs have the problems of structural instability and low conductivity [25], which limit their application in energy storage.

The combination of biochar and MOF materials can not only make up for the disadvantages of MOFs but also solve the problem of the structural collapse of materials to a certain extent [26] and help promote the coordination between the two, so they can play an important role in energy storage.

So far, biochar has been prepared by many methods, such as the template method [27] and pyrolysis [28]. Basically, the pore structure of the biochar prepared by the template method is orderly, and the pore size is uniform [18]. The traditional carbon-fired preparation process is simple. However, the template method is more complex and requires higher conditions. The pyrolysis method requires a high temperature environment, which may lead to adverse environmental impacts, including water pollution and greenhouse effect.

Therefore, this paper researched the composite materials of ZIF-67 and biochar. The methods of pyrolysis and an ionic liquid to compound with ZIF-67 material were systemically studied. The effects of the biochar preparation methods and the high temperature carbon materials on energy storage performance were successfully studied. The development of this work lays the foundation for biochar to be applied in electrochemical energy storage.

## 2. Experimental

### 2.1. Material Synthesis

#### 2.1.1. Preparation of Biochar

In this work, biochar was prepared by the ionic liquid method and the traditional carbon burning method. For the preparation of biochar by the ionic liquid method, an appropriate amount of clean and dry corn straw was firstly put into the reactor, and the ionic liquids lithium bromide (AR, Xiya reagent company, Chengdu, China), HCl (AR, Xiya reagent company, Chengdu, China), and dimethyl sulfoxide (AR, Xiya reagent company, Chengdu, China) were added according to the solid–liquid ratios of 1:10, 1:1.25, and 1:10, respectively. After adding the solution, we shook the reactor to ensure the solution was evenly distributed and sealed the reactor. The reaction kettle was put into the electric heating constant temperature blower drying box for the reaction, and we set the temperature to 140 °C for a duration of 170 min. The liquid components in the system were extracted by a filtration device at room temperature to obtain the filtration products. The product was washed twice with anhydrous ethanol. It was dried at room temperature to obtain the biochar. For the preparation of the biochar by pyrolysis, an appropriate amount of corn straw was put into the tubular furnace (SK-3-12-4), and N_2_ was introduced into the tubular furnace to drive out the air. The heating rate of the tubular furnace was 5 °C/min, the flow rate of N_2_ was 60 mL/min, and the final carbonization temperature was 800 °C. After heating to the final carbonization temperature, it remained at that temperature for 2 h; then, it was automatically cooled to room temperature to obtain the biochar.

#### 2.1.2. Preparation of ZIF-67

First, 5.82 g (20 mmol) of hexahydrate and cobalt nitrate was dissolved in 400 mL of methanol, and 6.56 g (80 mmol) of 2-methylimidazole was dissolved in 400 mL of methanol. Then, the cobalt nitrate hexahydrate solution was slowly added into the 2-methylimidazole solution by colloidal dropper. The solution became purple immediately after adding the cobalt nitrate hexahydrate. With the addition of the cobalt nitrate hexahydrate solution, a purple precipitate was gradually generated. The purple mixed solution was stirred in a magnetic stirrer at room temperature for 30 min; then, the reaction system settled for 12 h. After centrifugation, being washed, and being dried at 60 °C, the ZIF-67 samples were obtained.

#### 2.1.3. Fiber Biochar and ZIF-67 Composite

The ZIF-67 was combined with biochar by the in situ synthesis method. Firstly, three groups of cobalt nitrate hexahydrate solution were prepared, with 5.82 g (20 mmol) of hexahydrate and cobalt nitrate dissolved in 400 mL of methanol. In the three groups of prepared cobalt nitrate hexahydrate solutions, 0.5 g corn stalk, 0.5 g biochar prepared by ionic liquid, and 0.5 g biochar prepared by pyrolysis were added, respectively. The 2-methylimidazole solution was slowly added to the cobalt nitrate solution, and the precipitation was gradually generated with the addition of 2-methylimidazole. After complete addition, it was stirred at room temperature for 30 min and centrifuged to obtain the precipitation. After being dried at 60 °C, the product was placed in a porcelain cup and carbonized in a tube furnace at high temperature to obtain the composite product of the biochar and ZIF-67.

#### 2.1.4. Preparation of High-Temperature Charred ZIF-67 Material

The prepared ZIF-67 material was put into a tube furnace for carbonization at 800 °C to obtain the high-temperature charred ZIF-67 material.

### 2.2. Material Characterization

The morphology image of the biochar-ZIF-67 composites was obtained by Scanning Electron Microscopy (Philips XL-30S FEG, Los Angeles, CA, USA). The sample was placed on a conductive adhesive and was tested after being sprayed with gold. The material was analyzed by X-ray powder diffraction (XRD) using SmartLabSE, an X-ray diffractometer from Bruker, Berlin, Germany, to determine the structural properties and phase of the material in the range of 2θ (5–90). The scan rate was 20 min^−1^. The XRD test was performed at min^−1^ and a voltage of 40 kV.

### 2.3. Electrochemical Performance Test

Firstly, the electrode materials and capacitors were prepared, and cyclic voltammetry, an AC impedance test, and a constant current charge–discharge test were carried out on the materials by an electrochemical workstation.

In order to study the properties of the prepared biochar–MOF composites, several working electrodes were made of different materials, and the synthesized materials included ionic biochar ZIF-67, pyrolytic biochar-ZIF-67, corn stalk-ZIF-67, and high-temperature carbonization-ZIF-67. The materials were put into the pots, and an appropriate amount of N-methylpyrrolidone was dripped into the pots. The materials were ground carefully in the pots until a mud-like mixture was formed. The slurry-like mixture was evenly coated onto aluminum foil and scraped with a four-side preparation device to ensure the mixture was evenly distributed. The prepared two electrodes were taken as positive and negative electrodes, respectively. The electrodes, gaskets, and electrodes were placed in turn between the two glass slides and then clamped with clamps. The electrolyte KCl (1 M) was dropped into the gap of the capacitor clip to obtain a simple capacitor.

In the test system, the electrochemical workstation (Shanghai Chenhua Instrument Co., Ltd., model CHI660E, Shanghai, China) was used. The electrolyte was KCl (1 M), and the cyclic voltammetry (CV), electrochemical impedance spectroscopy (EIS) and constant current charge–discharge test (GCD) were performed. The specific capacity of the electrode materials was evaluated by a constant current charge and discharge test, and the calculation formula of the specific capacity was:(1)$$C_{m}=\frac{I△t}{m△V}$$
where cm (F·g^−1^) represents the specific capacity of the active material. △t(s) denotes the discharge time. I(A) denotes the discharge current. △V(V) represents the range of the discharge potential window. m(g) represents the quality of the active substances.

## 3. Results and Discussion

### 3.1. Properties of Charcoal-ZIF-67 Composite Material

#### 3.1.1. X-ray Powder Diffraction Studies

Figure 1a shows the XRD patterns of the ZIF-67 samples. Figure 1b shows the XRD diffraction pattern of the prepared ZIF-67 after carbonization. After carbonization at high temperature, all the original organic peaks of the ZIF-67 disappeared, and three obvious peaks different from the original peaks appeared, namely the characteristic peaks of cobalt. The characteristic peaks at 44.22°, 51.53°, and 75.87° corresponded to the crystal faces of (111), (200), and (220) of cobalt, respectively, which indicated that the chemical state of cobalt was transformed into cobalt nanoparticles after carbonization. Figure 1c shows the XRD patterns of the biochar–MOF composites prepared by the three different methods and the ZIF-67 after high temperature carbonization. The peaks of the ZIF-67 spectra of the three composites corresponded to those of pure carbon, indicating that the different biochar preparation methods had no effect on the structure and material composition of the ZIF-67. The four curves had characteristic peaks at 44.22°, 51.53°, and 75.87°, which corresponded to the (111), (200), and (220) crystal planes of cobalt, respectively. It can be seen from Figure 1 that the corresponding composite materials of the ionic biochar and pyrolytic biochar have sharp characteristic peaks, a small half-peak width, and better crystallinity and particle size.

#### 3.1.2. SEM Analysis

The structure and surface morphology of different samples were further analyzed by SEM. Figure 2a shows the microstructure of ZIF-67 obtained by the solvothermal method. The size of the ZIF-67 obtained by the solvothermal method was relatively uniform, and the particles showed a rhombic dodecahedron structure. The particle size was between 200 nm and 300 nm. Figure 2b shows the morphology of the products after the carbonization of ZIF-67. After high temperature carbonization, the original shape was basically maintained. During carbonization, the internal skeleton of ZIF-67 structure collapsed, and the cobalt element formed cobalt nanospheres, which meant that it lost its original ordered structure. Figure 2c shows the micromorphology of the straw and ZIF-67 composites, the biochar and ZIF-67 composites prepared by ionic liquid, and the biochar and ZIF-67 composites prepared by pyrolysis. The ZIF-67 particles were obviously and uniformly distributed and attached to the surface of the biochar. It can be seen that the biochar prepared by the straw and ionic liquid had a more uniform and complete adsorption form with the ZIF-67.

### 3.2. Electrochemical Performance Test for the Charcoal-ZIF-67 Composite Material

In the two-electrode system, the electrochemical behaviors of these materials were tested by cyclic voltammetry (CV), electrochemical impedance spectroscopy (EIS), and the constant current charge–discharge test (GCD). For the cyclic voltammetry test, Figure 3a–d shows the char burned ZIF-67, straw-ZIF-67, ion biochar-ZIF-67, and pyrolysis biochar-ZIF-67 under the scanning rates of 0.01, 0.05, 0.1, 0.5, and 1 V/s, electrolyte 1 mol/L KCl solution, cyclic voltammetry curves. Among them, the shape of the cyclic voltammetry curve of the charred ZIF-67 did not change with the increase in the scanning rate. The CV curves obtained at different scanning rates were similar to those obtained at rectangles. Even at high scanning rates, the CV curves displayed no obvious change, which indicated that the material had good electrochemical reversibility as electrode material.

Therefore, it showed a high diffusion rate and good pore structure, which is beneficial to ion transport. Figure 4 shows the charge–discharge curves of different materials at the current density of 0.01 A g^−1^. From Figure 4, it can be seen that there was no voltage sag when the constant current charge was changed into discharge, indicating that the series resistance of the electrode materials was small. It can be clearly seen that the specific capacitance of the ionic biochar-ZIF-67 composite was the largest at the same current density, and the specific capacitance was obtained from the charge–discharge curve, which was 15.6 F g^−1^, 63.54 F g^−1^, 18.75 F g^−1^, and 21.87 F g^−1^, respectively. In comparison, the specific capacitance was relatively low. Ma [29] showed that the special structure led to a higher specific capacitance (278 F g^−1^) and revealed that KOH could effectively increase the specific surface area of a novel shell biochar with an activation. Huang [30] showed that a calcination temperature of 750 °C demonstrated the highest specific capacitance, which was 201 F g^−1^ at a current density of 1 A g^−1^ in a 6 M KOH electrolyte. Qiao [31] prepared activated biochar material (PS-800-1000) by pyrolysis in molten KCl at 800 °C, and the heat treatment at 1000 °C exhibited excellent catalytic activity. The biochar materials reached a specific capacitance of 156 F g^−1^, and the biochar was obtained by the carbonization of spent malt rootlets with further processing by mild treatment in NaOH [32]. However, a high temperature and complicated process were needed in all of the abovementioned preparation methods of biochar.

Figure 5 shows the AC impedance diagram of different materials. For the capacitors without internal resistance, the EIS curve was generally a straight line; however, in practice, there will be internal resistance between the electrode and electrolyte of the capacitor, and the AC impedance diagram will appear semicircular in the high frequency region. Table 1 shows that the value of the ionic biochar-ZIF-67 material was relatively low, which confirmed the good performance of the ionic biochar-ZIF-67. Obviously, metal cobalt as an excellent electronic conductor had good conductivity. At the same time, the addition of carbon materials made the charge easier to transport between the active materials and the electrolyte.

## 4. Conclusions

In summary, three fiber biochar–MOF composites were successfully prepared in this research. According to the different preparation methods of the fiber biochar, the composites were applied to the electrode materials of supercapacitors. The electrochemical energy storage results showed that the maximum specific capacity of the biochar-ZIF-67 electrode prepared by the ionic liquid method was 63.54 F/g, which enhanced the performance of the fiber biochar electrochemical energy storage; furthermore, the electrode material had good electrochemical reversibility. The better performance of the supercapacitors can be attributed to the better morphology, crystallinity, and performance of the fiber biochar–MOF composites. The synthesis method can be easily expanded and provides another advantage for the production of such materials.

## Figures and Tables

**Figure 1 polymers-14-02419-f001:**
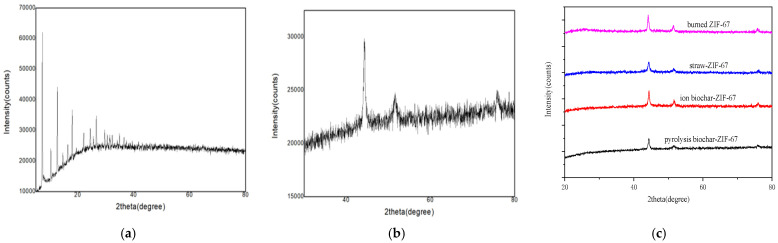
The XRD diffraction pattern of the prepared ZIF-67 sample (**a**), ZIF-67 after carbonization (**b**), and the biochar–MOF composites prepared by different methods (**c**).

**Figure 2 polymers-14-02419-f002:**
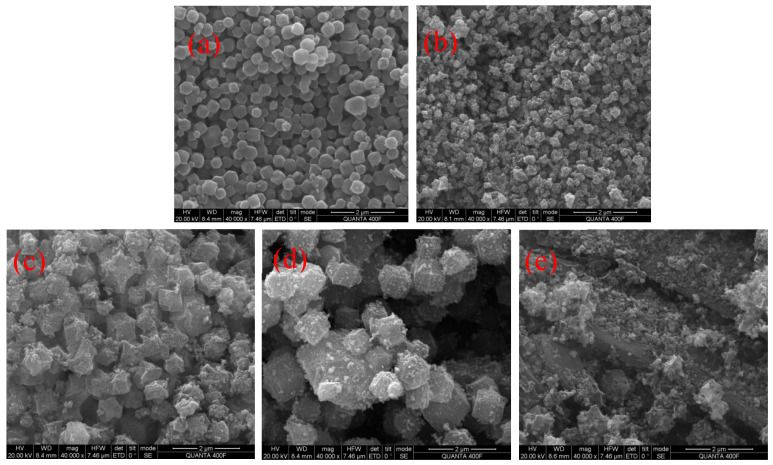
The SEM image of ZIF-67 (**a**), roasted ZIF-67 (**b**), straw-ZIF-67 composite (**c**), ionic liquid biochar-ZIF-67 composite (**d**), and pyrolysis biochar-ZIF-67 composite (**e**).

**Figure 3 polymers-14-02419-f003:**
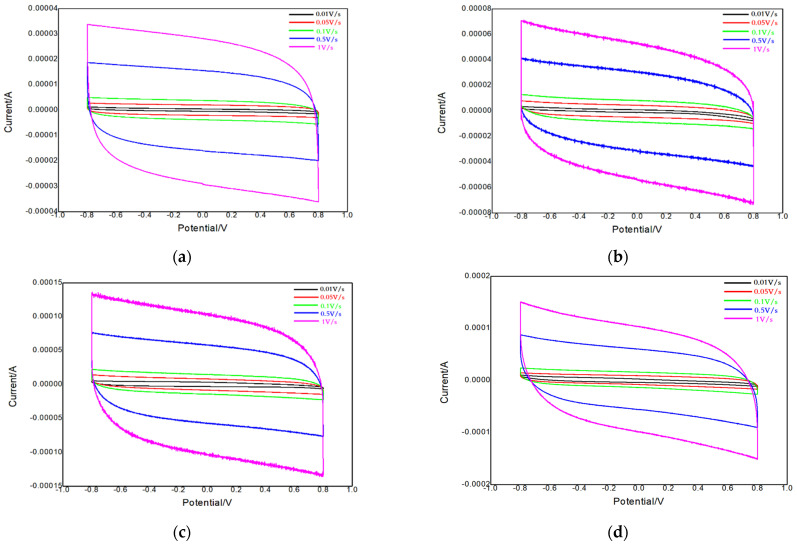
Cyclic voltammetry curves of the charred ZIF-67 (**a**), straw-ZIF-67 (**b**), ionic biochar-ZIF-67 (**c**), and pyrolytic biochar-ZIF-67 (**d**).

**Figure 4 polymers-14-02419-f004:**
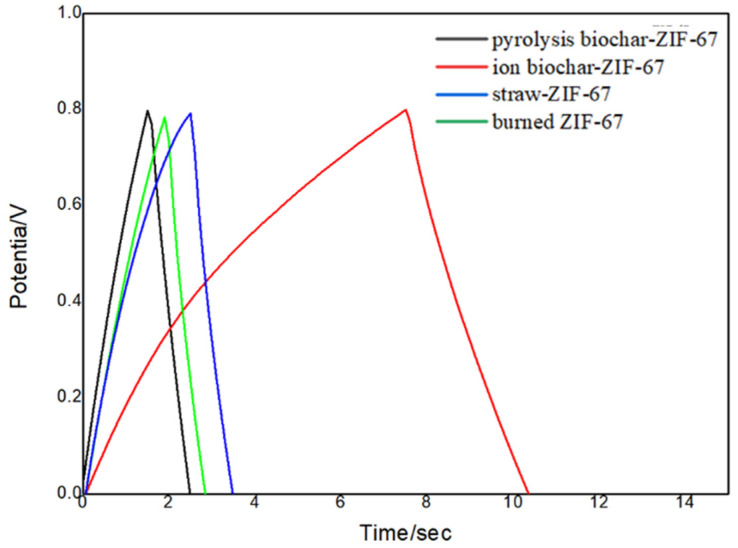
Charge–discharge curves of different materials at 0.01 A/g.

**Figure 5 polymers-14-02419-f005:**
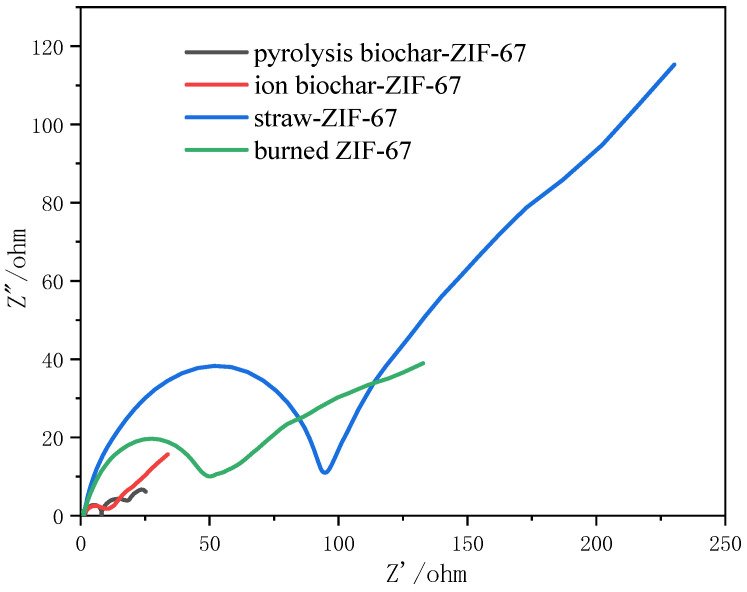
AC impedance diagrams for different materials.

**Table 1 polymers-14-02419-t001:** Values of the C impedance diagrams for different materials.

Sample	Values (ohm)
Pyrolysis biochar-ZIF-67	12
Ion biochar	10
Straw-ZIF-67	95
Burned ZIF-67	48

## Data Availability

Not applicable.

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
