# Peer review of "Application of Fiber Biochar–MOF Matrix Composites in Electrochemical Energy Storage"

_polymers, 2022, doi:10.3390/polym14122419_

Round 1

Reviewer 1 Report

The manuscript "Application of Fiber biochar-MOF matrix composites in Electrochemical energy storage" gives some interesting design and application of biochar for energy storage. Unfortunately, the manuscript itself has besides from many mistakes in text (English) need to be extensively checked also has several parts missing. The presentation is not qualified for a scientific paper

1. You present data but with no discussion to other works  with the References in numbers 1- 22 are far too few. Add meaningful discussion to your result parts with references to other work.

2. In the introduction there are experimental parts included which should be shown in experimental while in the introduction the goal why such work needed must be more clear formulated.

3. The abbreviation ZIF-67 appeared first in abstract and must be define there. Please modify it.

4. The presentation of figures are not well made its more like reading a student report than a scientific work. Not each figure need a separate number as speaking for Figure 1-3 can be placed as Figure 1a-c. 

5. Figure 4-6 SEM analysis are confusing also why there are 2 SEM images which are not explained either in capture nor in text. Please put those as one Figure together and name them a-e and add a figure capture (included scale bar) that gives the content of it.

6. Fig 7 -Fig 10. Cyclic voltammetry which electrolyte is applied and which reference is chosen. All those data need to be shown in Figure capture. Also those Figures as well can be merged in one Figure together. If you show current densities on Y axes those many decimals are confusing, either use scientific terms or use instead of A some as micro A.

7. The chronpotentiometric measurements  shown in Figure 11 (your labels are in chinese, please show English same counts for Figure 12) needs in the result parts a Table comparing your values in the text to other made in the field with similar compositions. Please add discussion as well a comparison. Additionally there no mean values shown so for scientific work reproducibility must be given.

8. The EIS measurements also lacks information, which electrolytes are applied and which material is shown as not evident from figure. Additionally if using impedance spectroscopy special fits (Randle´s equivalent circuit as example) are used to determine the ion conductivity. Please use such as well and show the ion conductivity which is important for capacitors in your work.

9. Additional experiments should be include for characterization of the composites such as FTIR and Raman spectroscopy as well TGA is needed. please use at least one of those to make your analysis more meaningful.

10. There are a lot of mistakes throughout the manuscript

Introduction: 

"..but there are problems such as small specific capacitance and small specific surface

area" please reformulate that.

"In addition, the yield of this method is lower than that ionic liquid method, and the reaction is not as thorough as ionic liquid method." Sentence confusing reformulate

Experimental. section 2.11: "ionic liquids LiBr, HCl and dimethyl sulfoxide" which ionic liquids are applied please specify as well list all chemicals where they purchased as well how they used (additional cleaning?).

"Remove the product after cooling the reactor". This sounds more like a protocol of someone given than a proper experimental description. Please reformulate

Same counts for : "Wash the purple precipitate twice with methanol, put the washed product into the oven for two hours"

2.1.4: high-temperature carbonization? Which temperature, which pressure is applied. There is nothing given in details. Please add all of it

2.2 "Surface Morphology of Biochar-ZIF-67 Composites by Scanning Electron Microscopy" that is no sentence please make extensive englisch correction (as a tipp show it to natural speaker).

There are a lot of those sentence where verb is missing or space or too many dots. Please correct those all.

Author Response

The manuscript "Application of Fiber biochar-MOF matrix composites in Electrochemical energy storage" gives some interesting design and application of biochar for energy storage. Unfortunately, the manuscript itself has besides from many mistakes in text (English) need to be extensively checked also has several parts missing. The presentation is not qualified for a scientific paper

  1. You present data but with no discussion to other works  with the References in numbers 1- 22 are far too few. Add meaningful discussion to your result parts with references to other work.

Response:Thank you for your advice, the comparisons with other studies have been supplied in the revise (in page 3).

  1. In the introduction there are experimental parts included which should be shown in experimental while in the introduction the goal why such work needed must be more clear formulated.

Response:Thank you for your advice, the introduction has been revised. (in page 3)

  1. The abbreviation ZIF-67 appeared first in abstract and must be define there. Please modify it.

Response:The ZIF-67 has been defined.

  1. The presentation of figures are not well made its more like reading a student report than a scientific work. Not each figure need a separate number as speaking for Figure 1-3 can be placed as Figure 1a-c. 

Response:The figures have been revised.

  1. Figure 4-6 SEM analysis are confusing also why there are 2 SEM images which are not explained either in capture nor in text. Please put those as one Figure together and name them a-e and add a figure capture (included scale bar) that gives the content of it.

Response:The figures have been revised.

  1. Fig 7 -Fig 10. Cyclic voltammetry which electrolyte is applied and which reference is chosen. All those data need to be shown in Figure capture. Also those Figures as well can be merged in one Figure together. If you show current densities on Y axes those many decimals are confusing, either use scientific terms or use instead of A some as micro A.

Response:The figures have been revised. The Y axes shows different densities because of the different mass of every samples.

  1. The chronpotentiometric measurements  shown in Figure 11 (your labels are in chinese, please show English same counts for Figure 12) needs in the result parts a Table comparing your values in the text to other made in the field with similar compositions. Please add discussion as well a comparison. Additionally there no mean values shown so for scientific work reproducibility must be given.

Response:The table has been added.

  1. The EIS measurements also lacks information, which electrolytes are applied and which material is shown as not evident from figure. Additionally if using impedance spectroscopy special fits (Randle´s equivalent circuit as example) are used to determine the ion conductivity. Please use such as well and show the ion conductivity which is important for capacitors in your work.

Response:We are sorry for that more works will proceed to do in next work.

  1. Additional experiments should be include for characterization of the composites such as FTIR and Raman spectroscopy as well TGA is needed. please use at least one of those to make your analysis more meaningful.

Response:We are very sorry that the FTIR, Raman and TGA have not done just yet.

  1. There are a lot of mistakes throughout the manuscript

Response:The English language of the article has been revised (marked in red).

Introduction: 

"..but there are problems such as small specific capacitance and small specific surface

area" please reformulate that.

"In addition, the yield of this method is lower than that ionic liquid method, and the reaction is not as thorough as ionic liquid method." Sentence confusing reformulate

Experimental. section 2.11: "ionic liquids LiBr, HCl and dimethyl sulfoxide" which ionic liquids are applied please specify as well list all chemicals where they purchased as well how they used (additional cleaning?).

"Remove the product after cooling the reactor". This sounds more like a protocol of someone given than a proper experimental description. Please reformulate

Same counts for : "Wash the purple precipitate twice with methanol, put the washed product into the oven for two hours"

2.1.4: high-temperature carbonization? Which temperature, which pressure is applied. There is nothing given in details. Please add all of it

2.2 "Surface Morphology of Biochar-ZIF-67 Composites by Scanning Electron Microscopy" that is no sentence please make extensive englisch correction (as a tipp show it to natural speaker).

There are a lot of those sentence where verb is missing or space or too many dots. Please correct those all.

Response:The sentence of the article has been revised (marked in red).

Reviewer 2 Report

In this manuscript, the authors prepared fiber biochar-MOF composites using different methods and investigated their morphology, crystallinity and electrochemical properties. I would suggest the acceptance of the manuscript after the following revision:

1. The introduction section needs some revision. I suggest the authors to add some background information about other biochar based composites, and point out why ZIF67 is superior to other materials. 

2. I suggest the authors to add more references. For example, some paper about the synthesis and application of MOF/ZIF67 can be cited: "Templated interfacial synthesis of metal-organic framework (MOF) nano-and micro-structures with precisely controlled shapes and sizes." Communications Chemistry 4.1 (2021): 1-10.; "Charge‐Separated and Lewis Paired Metal–Organic Framework for Anion Exchange and CO2 Chemical Fixation." Chemistry–A European Journal 26.61 (2020): 13788-13791.; "Comparative study between physicochemical characterization of biochar and metal organic frameworks (MOFs) as gas adsorbents." The Canadian Journal of Chemical Engineering 94.11 (2016): 2114-2120.

3. Section 2.2 material characterization, more detail should be given for SEM characterization, for example, how SEM samples were prepared, what's the accelerating voltage used. In addition, please double check if the XRD was performed at 4KV. 

4. Please make sure all figure legends are written in English (Fig.3 , Fig.11, Fig 12).

5. Please add scale bars in all SEM images.

6. In section 3.2, "It can be clearly seen from figure 11 that the specific capacitance of ionic biochar- ZIF-67 composite is the largest at the same current density" Why 0.01 A/g was chosen? Have the authors checked the performances of the material at other current density? 

7. "The better performance of supercapacitors attributes to better morphology, crystallinity and performance of fiber biochar-MOF composites." Please explain in more detail about how morphology and crystallinity are better for fiber biochar-MOF composites. XRD patterns in Fig 3 are quite similar for different composites. Also the authors mentioned in the manuscript "From figure 6, it can be seen that the morphology of ZIF-67 is complete and has good crystallinity." What does the "morphology of ZIF-67 is more complete" mean? How the degree of crystallinity was evaluated?

8. Please check typos and grammar error over the entire manuscript. Also, some sentences need to be rephrased. For example, last paragraph in the introduction section, "Therefore, the ZIF-67 material was first prepared in this study."  There are also unfinished sentences, for example "XRD test was performed at min-1 and voltage of 4 kV." 

Author Response

In this manuscript, the authors prepared fiber biochar-MOF composites using different methods and investigated their morphology, crystallinity and electrochemical properties. I would suggest the acceptance of the manuscript after the following revision:

  1. The introduction section needs some revision. I suggest the authors to add some background information about other biochar based composites, and point out why ZIF-67 is superior to other materials. 

Response:Thank you for your advice, some background information about other biochar composites has been added. (page in 3)

  1. I suggest the authors to add more references. For example, some paper about the synthesis and application of MOF/ZIF67 can be cited: "Templated interfacial synthesis of metal-organic framework (MOF) nano-and micro-structures with precisely controlled shapes and sizes." Communications Chemistry 4.1 (2021): 1-10.; "Charge‐Separated and Lewis Paired Metal–Organic Framework for Anion Exchange and CO2 Chemical Fixation." Chemistry–A European Journal 26.61 (2020): 13788-13791.; "Comparative study between physicochemical characterization of biochar and metal organic frameworks (MOFs) as gas adsorbents." The Canadian Journal of Chemical Engineering 94.11 (2016): 2114-2120.

Response:Thank you for your advice, the references have been added.

  1. Section 2.2 material characterization, more detail should be given for SEM characterization, for example, how SEM samples were prepared, what's the accelerating voltage used. In addition, please double check if the XRD was performed at 4KV. 

Response: The section about how SEM samples were prepared has been added. The XRD was performed at 40 kV.

  1. Please make sure all figure legends are written in English (Fig.3 , Fig.11, Fig 12).

Response:Thank you for your advice, all figures have been revised. 

  1. Please add scale bars in all SEM images.

Response:Thank you for your advice, the scale bars have been added.

  1. In section 3.2, "It can be clearly seen from figure 11 that the specific capacitance of ionic biochar- ZIF-67 composite is the largest at the same current density" Why 0.01 A/g was chosen? Have the authors checked the performances of the material at other current density? 

Response:The specific capacitance can be displayed very well in X minus Y coordinates under the 0.01A/g.

  1. "The better performance of supercapacitors attributes to better morphology, crystallinity and performance of fiber biochar-MOF composites." Please explain in more detail about how morphology and crystallinity are better for fiber biochar-MOF composites. XRD patterns in Fig 3 are quite similar for different composites. Also the authors mentioned in the manuscript "From figure 6, it can be seen that the morphology of ZIF-67 is complete and has good crystallinity." What does the "morphology of ZIF-67 is more complete" mean? How the degree of crystallinity was evaluated?

Response:It means the structure of ZIF-67 was not collapsed through calcinated. The four curves have characteristic peaks at 44.22 °, 51.53 ° and 75.87 °, which correspond to (111), (200) and (220) crystal planes of cobalt, respectively.

  1. Please check typos and grammar error over the entire manuscript. Also, some sentences need to be rephrased. For example, last paragraph in the introduction section, "Therefore, the ZIF-67 material was first prepared in this study."  There are also unfinished sentences, for example "XRD test was performed at min-1 and voltage of 4 kV." 

Response:The manuscript has been revised.   

Round 2

Reviewer 1 Report

The authors made revision but still there some major points that need to be addressed. There are still no discussion to other works included in the manuscript in the result parts. Also if reviewer ask add some additional experiments the answer we don't have done such is not sufficient. There need to be given a rebuttal.

Author Response

The manuscript "Application of Fiber biochar-MOF matrix composites in Electrochemical energy storage" gives some interesting design and application of biochar for energy storage. Unfortunately, the manuscript itself has besides from many mistakes in text (English) need to be extensively checked also has several parts missing. The presentation is not qualified for a scientific paper

  1. You present data but with no discussion to other works with the References in numbers 1- 22 are far too few. Add meaningful discussion to your result parts with references to other work.

Response:Thank you for your advice, the comparisons with other studies have been supplied in the revise (in page 3 and 10). The references are 32, now.

The modifcation methods mainly have a direct influence on the electrochemical performance of the rice biochar as the supercapacitor electrodes[20].

In comparison, the specific capacitance is relatively low. Ma[29] studied that the special structure lead to a higher specific capacitance (278 F/g) and revealed that KOH could effectively increase the specific surface area of biochar. novel shell biochar with an activation. Huang[30] researched that calcination temperature of 750 °C demonstrated the highest specific capacitance, which was 201 F g−1 at a current density of 1 A g−1 in a 6 M KOH electrolyte. Qiao[31] The prepared activated biochar material (PS-800-1000) by pyrolysis in molten KCl at 800 °C and heat-treatment at 1000 °C exhibits excellent catalytic activity. The biochar materials reached a specific capacitance of 156 F g−1 and biochar was obtained by the carbonization of spent malt rootlets with further processed by mild treatment in NaOH [32]. But the high temperature and complicated process were needed in all of the above-mentioned preparation methods of biochar.

[29] Ma Z W, Liu H Q, Lü Q F. Porous biochar derived from tea saponin for supercapacitor electrode: Effect of preparation technique. The Journal of Energy Storage, 2021, 40:102773.

[30] Huang S, Ding Y, Li Y, et al. Nitrogen and Sulfur Co-doped Hierarchical Porous Biochar Derived from the Pyrolysis of Mantis Shrimp Shell for Supercapacitor Electrodes. Energy & Fuels, 2021, 35(2):1557.

[31] Qiao Y, Zhang C, Kong F, et al. Activated biochar derived from peanut shells as the electrode materials with excellent performance in Zinc-air battery and supercapacitance. Waste Management, 2021, 125:257.

[32] Vakros J, Manariotis I D, Dracopoulos V, et al. Biochar from Spent Malt Rootlets and Its Application to an Energy Conversion and Storage Device. 2021, 9, 57.

  1. In the introduction there are experimental parts included which should be shown in experimental while in the introduction the goal why such work needed must be more clear formulated.

Response:Thank you for your advice, the introduction has been revised. (in page 3)

In this study, the ionic liquid method was adopted because it was very practical and responsive to the environment. In general, this room temperature-based synthesis method is to react with ionic liquids and other types of solutions respectively in a reactor. Specifically, the raw material was put into the drying oven at the temperature between 130 °C and 170 °C for about 170 min under high pressure. After cooling, the reaction material was filtered and dried to obtain biochar. This method not only keeps away from toxic organic solvents, but also saves energy waste and cost indirectly.

Therefore, this paper mainly researched the composite materials of ZIF-67 and biochar.   The prepared methods of pyrolysis and ionic liquid to compound with ZIF-67 material have been systemically studied. The electrochemical properties of different materials were tested by electrochemical workstation, and the effects of biochar preparation methods and high temperature carbon materials on energy storage performance were successfully studied. The development of this work laid the foundation for biochar applied in electrochemical energy storage.

  1. The abbreviation ZIF-67 appeared first in abstract and must be define there. Please modify it.

Response:The ZIF-67 has been defined.

Zeolite-type imidazolate skeleton -67 (ZIF-67) was revised in abstract.

  1. The presentation of figures are not well made its more like reading a student report than a scientific work. Not each figure need a separate number as speaking for Figure 1-3 can be placed as Figure 1a-c. 

Response:The figures have been revised.

(a)                     (b)                (c)

Fig. 1. XRD diffraction pattern of the prepared ZIF-67 sample (a), ZIF-67 after carbonization (b) and biochar-MOF composites prepared by different methods

  1. Figure 4-6 SEM analysis are confusing also why there are 2 SEM images which are not explained either in capture nor in text. Please put those as one Figure together and name them a-e and add a figure capture (included scale bar) that gives the content of it.

Response:The figures have been revised.

Fig. 4. The SEM image of ZIF-67 (a), roasted ZIF-67 (b), Straw-ZIF-67 composite (c), Ionic liquid biochar-ZIF-67 composite (d) and Pyrolysis biochar-ZIF-67 composite (e)

  1. Fig 7 -Fig 10. Cyclic voltammetry which electrolyte is applied and which reference is chosen. All those data need to be shown in Figure capture. Also those Figures as well can be merged in one Figure together. If you show current densities on Y axes those many decimals are confusing, either use scientific terms or use instead of A some as micro A.

Response:The figures have been revised. The Y axes shows different densities because of the different mass of every samples.

Fig. 5. Cyclic voltammetry curves of charred ZIF-67 (a), straw-ZIF-67 (b), ionic biochar-ZIF-67 (c) and pyrolytic biochar-ZIF-67 (d)

  1. The chronpotentiometric measurements shown in Figure 11 (your labels are in chinese, please show English same counts for Figure 12) needs in the result parts a Table comparing your values in the text to other made in the field with similar compositions. Please add discussion as well a comparison. Additionally there no mean values shown so for scientific work reproducibility must be given.

Response:The table has been added.

Fig. 6. Charge-discharge curves of different materials at 0.01 A/g

  1. The EIS measurements also lacks information, which electrolytes are applied and which material is shown as not evident from figure. Additionally if using impedance spectroscopy special fits (Randle´s equivalent circuit as example) are used to determine the ion conductivity. Please use such as well and show the ion conductivity which is important for capacitors in your work.

Response:We are sorry for that more works will proceed to do in next work.

Fig. 7. AC impedance diagrams for different materials

Table 1 Values of the C impedance diagrams for different materials

Sample

Values (ohm)

Pyrolysis biochar-ZIF-67

     12

Ion biochar

     10

Straw-ZIF-67

     95

Burned ZIF-67

     48

  1. Additional experiments should be include for characterization of the composites such as FTIR and Raman spectroscopy as well TGA is needed. please use at least one of those to make your analysis more meaningful.

Response:We are very sorry that the FTIR, Raman and TGA have not done just yet.

  1. There are a lot of mistakes throughout the manuscript

Response:The English language of the article has been revised (marked in red).

Introduction: 

"..but there are problems such as small specific capacitance and small specific surface

area" please reformulate that.

Response: revised: But small specific surface area limits its application in electrochemical energy storage.

"In addition, the yield of this method is lower than that ionic liquid method, and the reaction is not as thorough as ionic liquid method." Sentence confusing reformulate

Response: In addition, the yield of this method is lower than that ionic liquid method, and the reaction is not as thorough as ionic liquid method.

Experimental. section 2.11: "ionic liquids LiBr, HCl and dimethyl sulfoxide" which ionic liquids are applied please specify as well list all chemicals where they purchased as well how they used (additional cleaning?).

Response: revied: For the preparation of biochar by ionic liquid method, appropriate amount of clean and dry corn straw was first put into the reactor, and ionic liquids lithium bromide (LiBr, Xiya reagent company), HCl and dimethyl sulfoxide (Xiya reagent company) were added according to the solid-liquid ratios of 1: 10, 1: 1.25 and 1: 10, respectively.

"Remove the product after cooling the reactor". This sounds more like a protocol of someone given than a proper experimental description. Please reformulate

Response: After centrifugation, washed and dried at 60 °C, ZIF-67 samples were obtained.

Same counts for : "Wash the purple precipitate twice with methanol, put the washed product into the oven for two hours"

Response:  Wash and precipitate twice with methanol, put the washed product into the oven for two hours, set the temperature to.

After dried at 60 °C, the product was placed in a porcelain cup and carbonized in a tube furnace at high temperature to obtain the composite product of biochar and ZIF-67.

2.1.4: high-temperature carbonization? Which temperature, which pressure is applied. There is nothing given in details. Please add all of it

Response: The prepared ZIF-67 material was put into a tube furnace for carbonization at 800℃ to obtain high-temperature charred ZIF-67 material.

2.2 "Surface Morphology of Biochar-ZIF-67 Composites by Scanning Electron Microscopy" that is no sentence please make extensive englisch correction (as a tipp show it to natural speaker).

Response: The morphology image of biochar-ZIF-67 composites by Scanning Electron Microscopy. The sample sticks to a conductive adhesive and was texted after being sprayed gold. The material was analyzed by X-ray powder diffraction ( XRD ) using SmartLabSE, an X-ray diffractometer from Bruker, Germany, to determine the structural properties and phase of the material.In the range of 2θ(5-90)The scan rate is 20min-1. XRD test was performed at min −1 and voltage of 40 kV.

There are a lot of those sentence where verb is missing or space or too many dots. Please correct those all.

Response:The sentence of the article has been revised.

In this manuscript, the authors prepared fiber biochar-MOF composites using different methods and investigated their morphology, crystallinity and electrochemical properties. I would suggest the acceptance of the manuscript after the following revision:

  1. The introduction section needs some revision. I suggest the authors to add some background information about other biochar based composites, and point out why ZIF-67 is superior to other materials. 

Response:Thank you for your advice, some background information about other biochar composites has been added. (page in 3)

The modifcation methods mainly have a direct influence on the electrochemical performance of the rice biochar as the supercapacitor electrodes[20]. The single biochar is applied in the supercapacitor very few, so relatively simple process was adopted in this work that biochar was compounded with other materials to improve electrochemical performance. Other materials include organic semiconductors, inorganic metal complexes, organic metal complexes and else. Among these materials, metal-organic frameworks have become a good choice for supercapacitor applications due to their excellent physical and chemical properties and thermal stability[21].

  1. I suggest the authors to add more references. For example, some paper about the synthesis and application of MOF/ZIF67 can be cited: "Templated interfacial synthesis of metal-organic framework (MOF) nano-and micro-structures with precisely controlled shapes and sizes." Communications Chemistry 4.1 (2021): 1-10.; "Charge‐Separated and Lewis Paired Metal–Organic Framework for Anion Exchange and CO2 Chemical Fixation." Chemistry–A European Journal 26.61 (2020): 13788-13791.; "Comparative study between physicochemical characterization of biochar and metal organic frameworks (MOFs) as gas adsorbents." The Canadian Journal of Chemical Engineering 94.11 (2016): 2114-2120.

Response:Thank you for your advice, the references have been added.

MOFs, as a new type of porous material, have excellent characteristics of high porosity [7,8], large specific surface [9] and surface structure modification [10] compared with traditional energy storage materials. In recent years, synthesis and application of MOFs have attracted more and more attention[11-13]. This means that they have good microstructure and fast charge diffusion rate. ZIF-67 is one of the typical materials of cobalt-based MOFs, with adjustable pore size structure and good thermal stability[14].

  1. Section 2.2 material characterization, more detail should be given for SEM characterization, for example, how SEM samples were prepared, what's the accelerating voltage used. In addition, please double check if the XRD was performed at 4KV. 

Response: The section about how SEM samples were prepared has been added. The XRD was performed at 40 kV.

The morphology image of biochar-ZIF-67 composites by Scanning Electron Microscopy. The sample sticks to a conductive adhesive and was texted after being sprayed gold. The material was analyzed by X-ray powder diffraction ( XRD ) using SmartLabSE, an X-ray diffractometer from Bruker, Germany, to determine the structural properties and phase of the material.In the range of 2θ(5-90)The scan rate is 20min-1. XRD test was performed at min −1 and voltage of 40 kV.

  1. Please make sure all figure legends are written in English (Fig.3 , Fig.11, Fig 12).

Response:Thank you for your advice, all figures have been revised. 

  1. Please add scale bars in all SEM images.

Response:Thank you for your advice, the scale bars have been added.

Fig. 4. The SEM image of ZIF-67 (a), roasted ZIF-67 (b), Straw-ZIF-67 composite (c), Ionic liquid biochar-ZIF-67 composite (d) and Pyrolysis biochar-ZIF-67 composite (e)

  1. In section 3.2, "It can be clearly seen from figure 11 that the specific capacitance of ionic biochar- ZIF-67 composite is the largest at the same current density" Why 0.01 A/g was chosen? Have the authors checked the performances of the material at other current density? 

Response:The specific capacitance can be displayed very well in X minus Y coordinates under the 0.01A/g.

  1. "The better performance of supercapacitors attributes to better morphology, crystallinity and performance of fiber biochar-MOF composites." Please explain in more detail about how morphology and crystallinity are better for fiber biochar-MOF composites. XRD patterns in Fig 3 are quite similar for different composites. Also the authors mentioned in the manuscript "From figure 6, it can be seen that the morphology of ZIF-67 is complete and has good crystallinity." What does the "morphology of ZIF-67 is more complete" mean? How the degree of crystallinity was evaluated?

Response:It means the structure of ZIF-67 was not collapsed through calcinated. The four curves have characteristic peaks at 44.22 °, 51.53 ° and 75.87 °, which correspond to (111), (200) and (220) crystal planes of cobalt, respectively.

  1. Please check typos and grammar error over the entire manuscript. Also, some sentences need to be rephrased. For example, last paragraph in the introduction section, "Therefore, the ZIF-67 material was first prepared in this study."  There are also unfinished sentences, for example "XRD test was performed at min-1 and voltage of 4 kV." 

Response:The morphology image of biochar-ZIF-67 composites by Scanning Electron Microscopy. The sample sticks to a conductive adhesive and was texted after being sprayed gold. The material was analyzed by X-ray powder diffraction ( XRD ) using SmartLabSE, an X-ray diffractometer from Bruker, Germany, to determine the structural properties and phase of the material.In the range of 2θ(5-90)。The scan rate is 20。min-1. XRD test was performed at min −1 and voltage of 40 kV.
